# Minimally Invasive Surgical Approach in Granulomatosis with Polyangiitis Complicated with Intramural Descending Aorta Hematoma Followed by Aortic Wall Rupture

**DOI:** 10.3390/diagnostics15020144

**Published:** 2025-01-09

**Authors:** Mihai-Lucian Ciobica, Alexandru-Sebastian Botezatu, Zoltan Galajda, Mara Carsote, Claudiu Nistor, Bianca-Andreea Sandulescu

**Affiliations:** 1Department of Internal Medicine and Gastroenterology, “Carol Davila” University of Medicine and Pharmacy, 020021 Bucharest, Romania; lucian.ciobica@umfcd.ro (M.-L.C.); bianca-andreea.sandulescu@drd.umfcd.ro (B.-A.S.); 2Department of Internal Medicine I and Rheumatology, “Dr. Carol Davila” Central Military University Emergency Hospital, 010825 Bucharest, Romania; 3Department of Angiography and Cardiac Catheterization, “Dr. Carol Davila” Central Military Emergency University Hospital, 010825 Bucharest, Romania; sebastian.botezatu@gmail.com; 4Clinical Department of Cardiovascular Surgery, “Dr. Carol Davila” Central Military Emergency University Hospital, 010825 Bucharest, Romania; zgalajda@gmail.com; 5Department of Endocrinology, “Carol Davila” University of Medicine and Pharmacy, 020021 Bucharest, Romania; 6Department of Clinical Endocrinology V, “C.I. Parhon” National Institute of Endocrinology, 011863 Bucharest, Romania; 7Department 4-Cardio-Thoracic Pathology, Thoracic Surgery II Discipline, “Carol Davila” University of Medicine and Pharmacy, 050474 Bucharest, Romania; 8Thoracic Surgery Department, “Dr. Carol Davila” Central Military University Emergency Hospital, 010242 Bucharest, Romania; 9PhD Doctoral School, “Carol Davila” University of Medicine and Pharmacy, 020021 Bucharest, Romania

**Keywords:** autoimmune, granulomatosis with polyangiitis, Wegener’s granulomatosis, aortic rupture, minimally invasive surgery, aortic wall, hematoma, vertebral fracture, osteoporosis, glucocorticoids

## Abstract

**Background and Clinical Significance**: Granulomatosis with polyangiitis (GPA) represents a rare autoimmune disease with granulomatous inflammation, tissue necrosis, and systemic vasculitis of the small and medium blood vessels. Although the clinical elements vary, aortic involvement is exceptional and it represents a challenge that requires a rapid intervention with the potential of displaying a fulminant evolution. **Case Presentation**: We report a 64-year-old male with an 18-year history of GPA who presented atypical low back pain. Following ultrasound and computed tomography exams, the initial suspicion was an intramural descending aorta hematoma, surrounded by a peri-aortic sleeve suggesting a chronic inflammation. Serial non-invasive assessments revealed a progressive lesion within the next 10 to 12 days to an aortic wall rupture, despite the absence of previous aneurysmal changes. The peri-aortic fibrous inflammatory sleeve was life-saving, and emergency minimally invasive surgery was successful, including the massive improvement in back pain. **Conclusions**: To our knowledge, this is a very rare scenario in GPA; we found only 18 other cases (the oldest report being from 1994). An interventional approach was mentioned in a few cases as seen in this instance. Glucocorticoid medication for GPA might act as a potential contributor to symptomatic osteoporotic fractures which require a prompt differential diagnosis. Unusual aortic manifestations (such as intramural aortic hematoma or aortic wall rupture) are difficult to recognize since the index of clinical suspicion is rather low. A prompt intervention may be life-saving and a multidisciplinary team is mandatory. Minimally invasive surgical correction of the aortic event represents an optimum management in the modern era. Such cases add to the limited data we have so far with respect to unusual outcomes in long-standing GPAs.

## 1. Introduction

Granulomatosis with polyangiitis (GPA) (prior known as Wegener’s granulomatosis) represents a rare autoimmune disease characterized by a systemic necrotizing granulomatous vasculitis of small- and medium-sized blood vessels in association with positive cytoplasmic-staining anti-neutrophil cytoplasmic antibodies (cANCA) [1]. The presence of cANCA directed against proteinase 3 (PR3-ANCA) supports the diagnosis and it is highly specific for GPA. The latest data showed that the use of both immunofluorescence and ELISA enhances the sensitivity and specificity of the diagnosis of an ANCA-associated vasculitis to 96% and 98.5%, respectively. Some patients with GPA express myeloperoxidase perinuclear-staining ANCA (p-ANCA) specific for myeloperoxidase (MPO-ANCA) and less than 10% of the cases have no detectable ANCA [1,2,3,4,5,6,7,8,9].

Approximately 20% of the patients are seronegative; thus, the diagnosis and the treatment may be delayed [2]. The pathogenesis remains unknown, it appears to be multifactorial, involving the interaction of genetically predisposed individuals with different environmental triggers [1,3]. Some authors suggested an ethnic and geographic susceptibility, with a higher incidence in Europe (especially in the Northern area) and Australia compared to the Asian and American countries [4,5,6,7,8,9,10]. In Europe, the overall incidence was found between 0.5 and 20 subjects per million population per year (a prevalence varying from 20 to 160 cases per million per year) [5,6,7,8,9,10,11,12]. Males are more prone than females (an incidence of 2.25 per 100,000 in men versus 1.63 per 100,000 in women was identified), but some data suggested a similar gender-related rate [5,6,7,8,9,10,11,12,13,14,15,16,17,18,19]. The peak age at diagnosis is at 60–65 years [20,21].

Regarding the pathogenic findings in GPA, the inflammation itself mainly affects the blood vessels of the upper respiratory tract, lungs, and kidneys [2]. Relapses are common and long-term complications are induced not only by the disease, but also by the medication. The 10-year survival rate varies from 40% to 70%, with renal damage being the most important contributor to the overall disease burden and mortality in GPA [1,22,23,24]. Generally, the pathogenic mechanisms of GPA remain less understood, with most studies in this field suggesting an interplay between genetic and environmental factors that leads to the clinical manifestations of the disease and development of ANCA autoantibodies (that have been proven to be pathogenic in both in vitro and in vivo studies) [1,25,26,27,28,29,30,31]. Genome-wide association studies showed single nucleotide polymorphism in human leucocyte antigen (HLA) regions such as HLA-DP and non-HLA regions (e.g., the gene encoding alpha 1-antitrypsin) that might play a role in this specific matter [32,33,34,35,36,37,38,39,40,41,42,43,44,45].

The clinical presentation is highly variable, although respiratory and renal manifestations are most commonly observed in daily practice [3]. Non-specific systemic manifestations such as the loss of appetite (leading to weight loss), fever, night sweats, myalgia, etc. may be identified in half of the GPA cases. On the other hand, the granulomatous inflammation of the upper respiratory tract, without systemic symptoms, has been described in a third of the patients. Another form of presentation is a rapidly progressive disease from onset in association with multiple organ involvement. The inflammatory aspects are found at respiratory, renal, auricular, nasal, pharyngeal, and ocular levels. The lesions of the respiratory tract include lung nodules, cavitating lesions, and pulmonary capillaritis with alveolar hemorrhage, and the associated clinical manifestations include shortness of breath, cough, and hemoptysis. Renal involvement may be asymptomatic, but laboratory assessments identify hematuria, proteinuria, and increased levels of serum creatinine, while pauci-immune crescentic glomerulonephritis is confirmed via kidney biopsy. Otolaryngology issues include nasal crusts, polyps, mucosal ulcerations leading to nasal hemorrhage, nasal cartilage destruction with saddle nose deformity, otitis media, loss of hearing as well as rhinitis, sinusitis, and laryngitis with chronic evolution. A life-threatening manifestation of GPA is subglottic stenosis. Ocular involvement involves conjunctivitis, keratitis, episcleritis, and uveitis; alternatively, the formation of orbital and retro-orbital granulomatous masses causes pain, proptosis, and diplopia. Cutaneous lesions are a consequence of dermal capillary inflammation, causing purpura, petechiae, or even skin ulcerations [25,26,27,28,29,30,31].

The rate of large vessel involvement in GPA, especially aortic or peri-aortic complications, is extremely low (0.1 cases per million), ranging from an incidental finding in a completely asymptomatic subject with respect to this specific issue to a very severe and dramatic picture which includes aortic dissection or rupture [16,23]. The underlying mechanisms of ANCA-related aortitis are similar to those found in small vessel damage [24], while the approach requires prompt recognition and a multidisciplinary strategy.

### Objective

Our purpose was to introduce an exceptional case of aortic rupture that has been successfully managed amid minimally invasive surgical techniques in a patient confirmed with GPA that is associated with multiple complications, including symptomatic vertebral fractures caused by glucocorticoid-related osteoporosis; the atypical low back pain represented the initial index of suspicion for further investigations which allowed a prompt diagnosis of the aortitis.

## 2. Methods

In addition to the presentation of the case-associated clinical, laboratory, imaging, and histological analyses, we provided in Discussion Section a review of the literature with concern to this specific topic. Specifically, we performed a PubMed-based, English literature search from Inception until July 2024 by using the key terms “granulomatosis with polyangiitis” (alternatively, “Wegener’s granulomatosis’”) and “aorta” (alternatively, “aortitis”) in different combinations. We selected original reports that met both of the following criteria: clinical case confirmed with ANCA-positive GPA (or GPA confirmed based on highly specific histological features) in addition to the identification of an aortic involvement that had been demonstrated by imaging and/or pathological exams and assessed as being part of the GPA-related picture. We identified 21 such cases and among them, two non-English publications and another without detailed clinical information were excluded, hence resulting in 18 similar reports [46,47,48,49,50,51,52,53,54,55,56,57,58,59,60,61,62,63] prior to this vignette on point (Figure 1).

## 3. Case Presentation

This was a 64-year-old male patient with an 18-year history of GPA who presented progressively increased lumbar pain over the last month.

### 3.1. Medical History

The medical history was notable for chronic hepatitis B infection and hypertension since his early 40s. The clinical onset of the GPA included fever, poly-arthralgia (at the level of knees, ankles, and elbows), skin rash, cough, and hemoptysis. Serological lab tests revealed inflammatory syndrome, leukocytosis, moderate anemia, and elevated anti-neutrophil cytoplasmic antibody titers with positive reactions against proteinase-3 (PR3-cANCA) (Appendix A). The chest X-ray showed an alveolar infiltrate in the middle lobe of the right lung (Figure 2).

At the time, the pathological examination via biopsy specimen from the nasal cavity confirmed GPA (Figure 3).

Subsequent renal damage was stabilized after six months of induction treatment with cyclophosphamide (initially intravenously pulses, then orally) and corticosteroids, followed by maintenance therapy with azathioprine. Favorable evolution was also confirmed via nasal biopsy re-examination (Figure 4).

The disease remained in remission under therapy for seven years followed by a significant clinical relapse (systemic manifestations such as fever and bilateral pulmonary infiltrates) (Figure 5).

At that point, the condition was controlled with glucocorticoids and azathioprine. This episode was followed by a long-term evolution without exacerbations, but successively involved other co-morbidities including chronic kidney disease (estimated glomerular filtration rate between 50 and 70 mL/min/1.73 m^2^). Moreover, chronic hepatitis B was treated with entecavir. He developed aseptic necrosis of the left femoral head that required total hip arthroplasty (eleven years since the initial GPA diagnosis), but, also, aortic valve damage in terms of degenerative aortic stenosis that required valve prosthesis (sixteen years since first GPA identification). Additionally, he experienced multiple vertebral fractures (that required prolonged analgesic treatment and a lumbar corset). In the meantime, he was offered to continue with azathioprine (50 mg per day).

### 3.2. Current Hospitalization

#### 3.2.1. A Fine Index of Clinical Suspicion: Low Back Pain

At current admission, the physical examination highlighted the (spontaneous) low back pain that was increased by mobilization and prolonged orthostatism, and it became slightly improved at bed rest, with anterior irradiation to the hypogastrium. The recurrence of pulmonary manifestations occurred approximately 5 years after the discontinuation of the oral cyclophosphamide therapy (which had been maintained for about 2 years). At that time, the patient was on low-dose maintenance corticosteroid therapy (7.5–10 mg/day). Subsequently, azathioprine was introduced as a maintenance remission therapy. In addition, laboratory tests showed immunological anomalies (positive cANCA), a relatively stable serum creatinine as seen during recent years (stage 3 chronic kidney disease), and inflammatory syndrome (C-reactive protein of 125 mg/L).

There were no consistent elements of GPA reactivation on the chest X-ray while a lumbar profile spine X-ray was suggestive of osteoporosis with multiple vertebral fractures (most likely, a glucocorticoid-induced osteoporosis) (Figure 6).

Noting the clinical presentation and the results of the mentioned investigations, the preliminary diagnosis was an atypical low back pain and the prior osteoporotic vertebral fractures did not seem enough to explain the unusual clinical traits. The patient was initially treated with parenteral analgesics, myorelaxants, and bed rest, with a mild initial clinical improvement. Hypotensive and anticoagulant background therapy was continued (due to the metal prosthesis of the aortic valve). However, alternative (non-vertebral-related) diagnoses were taken into consideration such as mesenteric ischemia or a retroperitoneal condition, and thus, while continuing the pain medication, further exams were performed.

#### 3.2.2. Additional Imaging Assessments

Abdominal ultrasound showed no significant anomalies of the intra-abdominal organs, except for gallstones and a hypoechoic area at the aortic wall (that had a thickness of 0.2–0.3 cm with a crescent shape), located adjacent to the emergence of the inferior mesenteric artery and embedded in a significant retroperitoneal fibrous thickening.

Further on, a computed tomography (CT) scan (with contrast) was performed and confirmed an aortic parietal lesion surrounded by retroperitoneal fibrosis without other significant pathological elements (Appendix B) (Figure 7).

A multidisciplinary assessment was performed and ruled out a GPA-related aortic dissection, thus a conservative approach with symptomatic treatment was followed. The patient initially received analgesic treatment with acetaminophen and metamizole sodium, but with partial relief of symptoms that required the subsequent escalation of therapy to nefopam hydrochloride and tramadol. Also, special attention was directed to achieving strict blood pressure control. However, close surveillance amid continuing the current hospitalization was provided while the evolution was undulating, with episodes of clinical improvement alternating with exacerbations of the low back pain with the same pattern and a suboptimal control under analgesics for more than 10 days.

#### 3.2.3. The Descending Aorta Rupture

While continuing the symptomatic treatment, a serial abdominal ultrasound was performed noting the initial aortic features. After 12 days of hospital stay, the pain became very severe with an altered general status (associated with sweating and low blood pressure). This time, abdominal ultrasound revealed an enlargement of the mentioned hypoechoic para-aortic area, surrounded by a hyperechoic area of inflammatory tissue (Figure 8).

The location of the blood extravasation was identified by using the color flow Doppler exam (Figure 9).

Emergency CT re-scan confirmed an intramural aortic hematoma (two-thirds of its circumference) of 3 cm (cranio-caudal plane) with a 1.5 cm thickness (Figure 10).

In this context, we correlated the persistent low back pain with this identified aortic lesion. The patient was once again re-assessed by a multidisciplinary team and transferred to the Department of Vascular Surgery.

#### 3.2.4. Minimally Invasive Surgery

A minimally invasive emergency surgical approach was taken after administering fresh frozen plasma and cryoprecipitate transfusions (the patient was under anticoagulant treatment). Pre-interventional arteriography highlighted the abdominal aorta rupture and the formation of a pseudoaneurysm with a maximum diameter of 2 by 1 cm (Figure 11).

The endovascular surgery was performed via a femoral artery approach. Two endovascular prostheses with a diameter of 2.4 by 4 cm were implanted after balloon dilation. It should be noted that the aortic wall appeared weak and thin (most likely an effect of chronic inflammation). The control injection showed a complete occlusion of the aortic rupture and the absence of the pseudo-aneurysm loading (Figure 12).

The peri-aortic fibrous inflammatory sleeve was a life-saving and minimally invasive surgery that was performed in time as an emergency procedure that contributed to the success of the approach, with a good evolution and rapid improvement in the symptoms, including the remission of the low back pain.

#### 3.2.5. One-Month Post-Surgery Follow-Up

The ultrasound scan performed one month later showed a regression of the aortic parietal lesion, without flow on the Doppler examination. The clinical and biological status normalized (Figure 13). Life-long follow-up is mandatory amid the primary condition and the mentioned co-morbidities.

## 4. Discussion

### 4.1. Case-Focused Analysis

In this vignette, we introduce an exceptional complication of GPA, namely, an intramural aortic hematoma complicated with an aortic wall rupture in a patient who otherwise had multiple co-morbidities, including symptomatic vertebral fractures that potentially misled or delayed the diagnosis of the aortic involvement with concern to the presence of severe and atypical low back pain. Numerous (non-GPA-related) contributors may be co-present in one individual and cause an aortic rupture such as cardio-metabolic, oncologic, or endocrine ailments, but, also, autoimmune disorders, namely, large vessels vasculitis like giant cell arteritis and Takayasu arteritis [64,65,66,67,68].

Imaging techniques for close surveillance, specifically, ultrasound as seen in this case, are effective for an early identification of an aortic complication under special circumstances, thus offering a good prognosis [68,69,70,71,72]. Additionally, we opted for CT, but alternative tools (e.g., magnetic resonance imaging or positron emission tomography–CT) might play a crucial role in the overall management [68,72]. Some co-morbidity might selectively limit an excessive use of these modern tools such as the presence of chronic kidney disease or orthopedic prosthesis [71,72] as found in this 64-year-old male.

In this case, a minimally invasive surgical procedure prevented a fatal outcome. Generally, patients confirmed with chronic renal failure are prone to more side effects and a lower rate of success with respect to post-operatory cardiovascular morbidity and mortality due to the presence of an already impaired endothelium; nevertheless, they should benefit from the procedures and an early intervention allows a better outcome [73,74].

In this instance, the clinical index of suspicion that has been provided by the atypical low back pain became a powerful tool in addition to the serial abdominal ultrasound exams. Acute painful osteoporotic vertebral fractures represent a major cause of reduced mobility, functionality, and quality of life, including in glucocorticoid-derivate osteoporosis. The conservative approach includes long-term rehabilitation procedures to control the pain in association with medication such as non-steroidal anti-inflammatory drugs as well as calcitonin in the short term, and bisphosphonates or teriparatide in the long term which are not applicable in the case of chronic kidney disease [75,76,77,78]. Overall, the multimodal management of vertebral fractures, the most common osteoporotic fractures in daily practice, is less than ideal, requiring a personalized approach, while more than one-third of the patients will remain symptomatic lifelong in terms of chronic pain, neurologic impairment, spine deformity, or chronic pseudarthrosis (even after some of them became surgery candidates) [79,80,81].

### 4.2. Sample-Focused Analysis

As mentioned, we found 18 other cases with aortic involvement in GPA according to our mentioned methods of search [46,47,48,49,50,51,52,53,54,55,56,57,58,59,60,61,62,63] (Table 1).

These 18 cases [46,47,48,49,50,51,52,53,54,55,56,57,58,59,60,61,62,63] (the oldest report being from 1994 [63]) revealed an overall good outcome, except for one 63-year-old male with fatal evolution due to a circulatory collapse [61]. An interventional approach was specifically mentioned in five cases [50,55,58,60,62] as seen in the present vignette. The sex ratio showed a clear male predominance (male–female ratio of 3.5), with ages at aortic involvement recognition varying between 28 and 74 years in the male cohort (*N* = 14 subjects, average age of 52.5), respectively, between 60 and 71 years in the female group (*N* = 4 patients, mean age of 65). Only three individuals [49,50,55] were younger than 40 years (16.66%). The aorta was affected at any anatomic level with different types of ailments (including combinations) such as aortitis, thrombus, hematoma, aortic dissection, aneurysm, peri-aortitis, etc. [46,47,48,49,50,51,52,53,54,55,56,57,58,59,60,61,62,63,64,65,66,67,68,69,70,71,72,73].

The heterogeneous clinical spectrum at presentation included the following: chest, thoracic back, or low abdominal pain; vomiting, hemoptysis; weight loss; dyspnea; cough; facial swelling; nasal congestion; headache; loss of consciousness; hematuria; paresthesia; and peripheral edema [46,47,48,49,50,51,52,53,54,55,56,57,58,59,60,61,62,63]. Similarly to the present case, we identified one admission complicated with excessive sweats [48], another two patients with fever [54,56], and other three reports manifested with malaise [56,58,63]. Remarkably, three men showed low back pain, aged 38 [55], 63 [60], and 74 [53], as found in this 64-year-old male case. Of note, glucocorticoid medication for GPA was identified in the medical history of the majority of these cases as a potential contributor to the symptomatic osteoporotic fractures which required a prompt differential diagnosis [46,47,48,49,50,51,52,53,54,55,56,57,58,59,60,61,62,63].

As shown, our case findings are in line with other similar studies. The emergency according to the clinical presentation and the minimally invasive type of intervention did not allow performing a biopsy, but the peri-aortic inflammatory tissue provided consistent data as seen in other cases [50,55,58,60,62] whereas the autopsy or the surgical biopsies that have been taken from the aorta and the retroperitoneal inflammatory tissue showed granulomatous vasculitis.

Of note, the patient on point had some non-modifiable traditional cardiovascular risk factors (gender and age) in addition to arterial hypertension, a modifiable risk factor that was permanently under adequate treatment with optimum results in this specific instance. Nevertheless, it should be mentioned that in recent years, multiple studies, including large randomized control trials, have shown that atherosclerosis cannot be explained only by traditional risk factors, but also by the existence of an autoimmune process that plays an important role in the pathogenesis of the associated lesions, the risk of cardiovascular events in patients diagnosed with ANCA-associated vasculitis being described as 65% higher when compared with the general population. Disease-specific factors, as pointed out by the Birmingham Vasculitis Activity Score and C-reactive protein levels, might indicate an intense systemic inflammation as a particular contributor to accelerating the atherosclerotic processes [82,83,84,85,86,87,88]. Generally, small vessel vasculitis is associated with an increased level of pro-inflammatory cytokines that induce endothelial damage as well as inflammation and necrosis that lead to ischemia, and sometimes an occlusion of the blood vessels is registered. Atherosclerosis and ANCA-associated vasculitis share some molecular pathogenic mechanisms. Both conditions result in arterial stiffening while a reduction in the endothelium-dependent vasodilatation is determined by an over-production of the inflammatory cytokines and the reactive oxygen species due to the activation of the mononuclear phagocytes and complement system. These amplify the endothelial destruction and increase the circulating low-density lipoprotein-cholesterol and endothelin-1 production by the endothelial cells, both with pro-thrombotic effects [82,83,84,85,86,87,88,89].

Among the limitations of this case study, we mention the real-life setting and, hence, the generalizability is not elevated, noting a small number of similar case reports as we could identify them according to the mentioned methods. Moreover, long-term follow-up data are mandatory to highlight the impact of the minimally invasive surgical procedure on the overall long-standing outcome in addition to the complex picture of complications and co-morbidities. As a practical point, the clinical presentation was less suggestive in this instance, while the association with multiple co-morbidities (e.g., chronic kidney disease) and the distinct injury site (aorta) made it difficult to perform some additional investigations (such as using contrast-based enhanced imaging assessment or performing a vascular biopsy). While a PET-CT scan might prove useful to reveal some inflammatory changes in the aortic wall, in this case, the initial and short-term evaluation via ultrasound, a widely available tool, allowed the aortic lesion recognition and offered an optimum time frame of intervention. Moreover, PR3-ANCA was tested at the time of diagnosis (the values were 10 times above upper normal limits) and afterward (and they showed a slightly elevated level). Although PR3-ANCA is a consistent biomarker for GPA diagnosis, the risk of relapse is not specifically predicted by PR3-ANCA levels alone [88]. At the time of that admission, the laboratory tests showed a significant inflammatory syndrome (C-reactive protein of 125 mg/dL, and an erythrocyte sedimentation rate of 97 mm/h). It should be noted that other causes of systemic inflammation (e.g., infectious or neoplastic) were excluded from the first days of presentation. We took into consideration performing a PET-CT scan, but the rapid progression of the aortic lesion required an immediate intervention (hence, there was no time to prove the inflammatory nature of the phenomenon). Further on, the impact of glucocorticoid exposure on developing this aortic involvement remains an open issue. Additional management challenges are expected in relationship with the abnormal kidney function and the decision of dialysis as well as choosing the best option therapy for glucocorticoid osteoporosis.

## 5. Conclusions

To summarize the key messages of this case introducing a 64-year-old male with an 18-year GPA history complicated with aortic involvement and associated review of the literature:Unusual aortic manifestations (such as intramural aortic hematoma followed by an aortic rupture) of this rare disease are difficult to recognize since the index of clinical suspicion is rather low;Prompt intervention may be life-saving and a multidisciplinary team is mandatory;Minimally invasive surgical correction of the aortic event represents optimum management in the modern era;This case adds to the limited series of similar reports in the literature (18 prior cases) with a mean age at diagnosis of 55.27 years, and a male–female ratio of 3.5;Atypical low back pain in a patient with glucocorticoid-induced osteoporosis might not always be a symptomatic vertebral fracture.

## Figures and Tables

**Figure 1 diagnostics-15-00144-f001:**
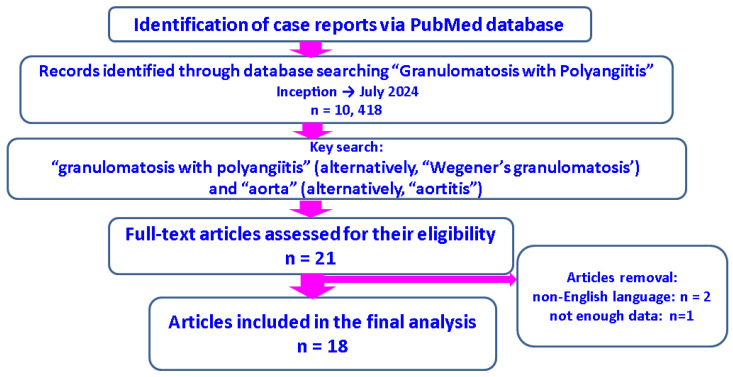
Flow diagram of literature search according to our methods [46,47,48,49,50,51,52,53,54,55,56,57,58,59,60,61,62,63].

**Figure 2 diagnostics-15-00144-f002:**
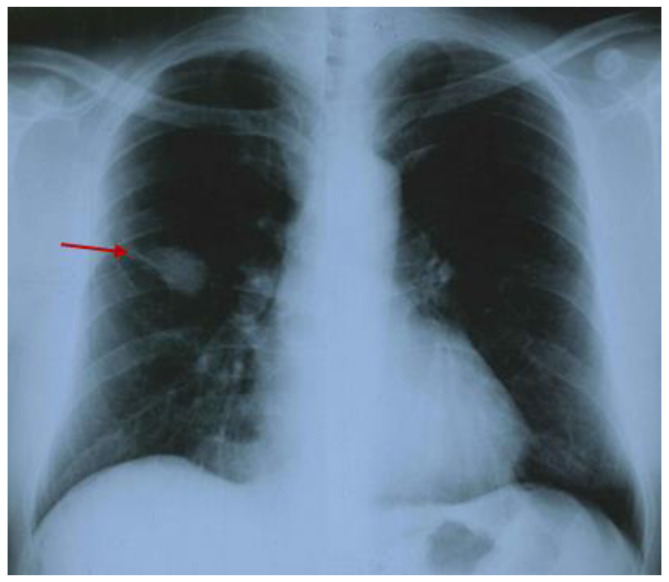
Chest X-ray showing an alveolar infiltrate of 3 by 4 cm (red arrow) in the middle lobe of the right lung.

**Figure 3 diagnostics-15-00144-f003:**
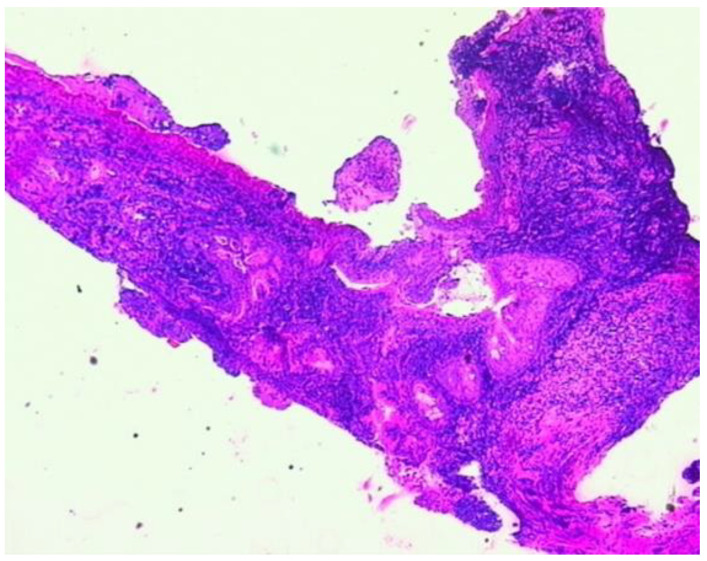
Histological examination (hematoxylin–eosin) via nasal tissue biopsy showing necrotizing vasculitis, inflammatory cell infiltration, and granulomatous reaction which represent suggestive features for GPA (magnification ×5).

**Figure 4 diagnostics-15-00144-f004:**
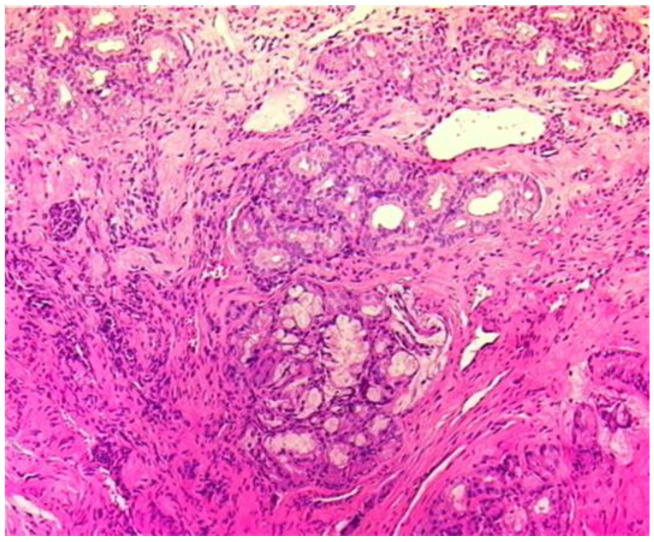
Histological examination (hematoxylin–eosin) via nasal tissue biopsy after six months of treatment revealing mucinous glandular structures with preserved architecture, moderate lympho-plasmocytic inflammatory infiltrates (with diffuse distribution), moderate interstitial fibrosis, and a few dilated capillary vessels (magnification ×20).

**Figure 5 diagnostics-15-00144-f005:**
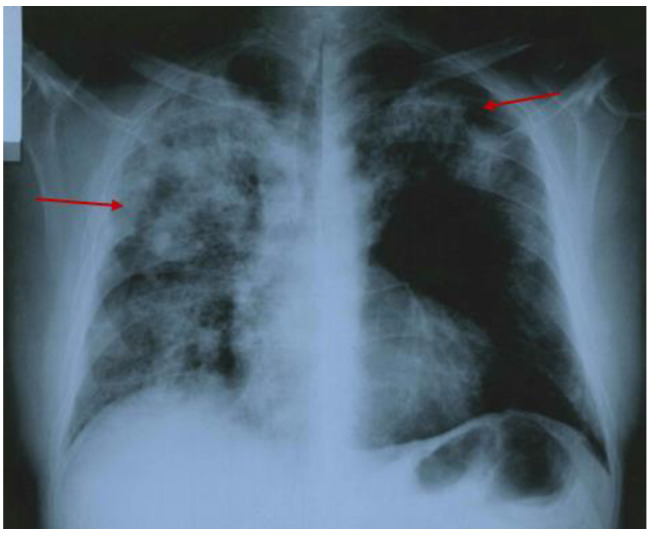
Chest X-ray showing diffuse bilateral pulmonary infiltrates (red arrow) predominantly affecting the right lung seven years after prior GPA diagnosis.

**Figure 6 diagnostics-15-00144-f006:**
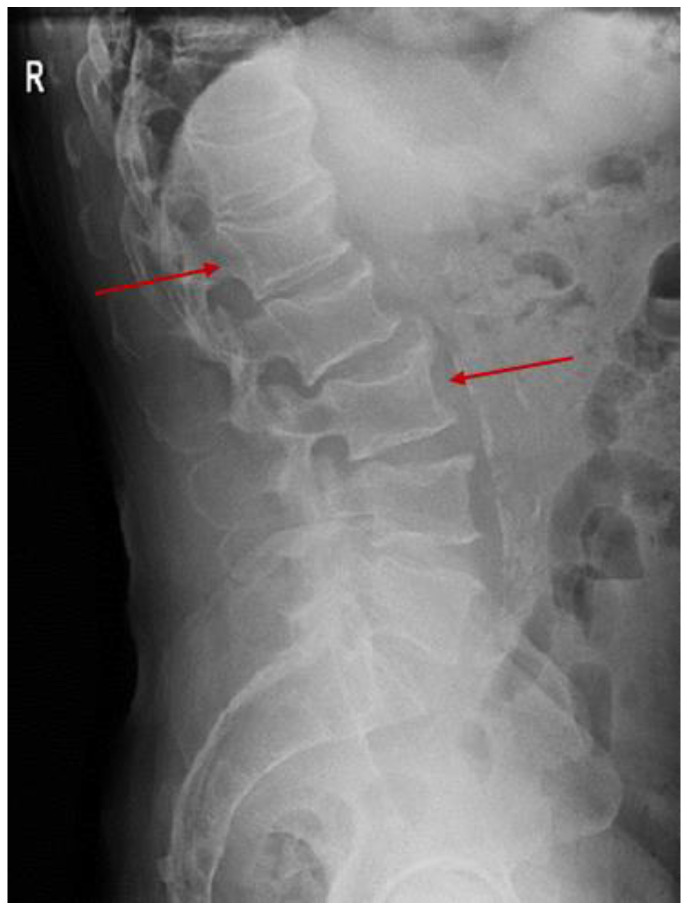
Profile X-ray of the spine showing anomalies of the vertebral body’s shape and size with a reduction in the lumbar L1 vertebral body height and concave vertebral deformities—predominantly at lumbar L1 and L3 level (red arrows).

**Figure 7 diagnostics-15-00144-f007:**
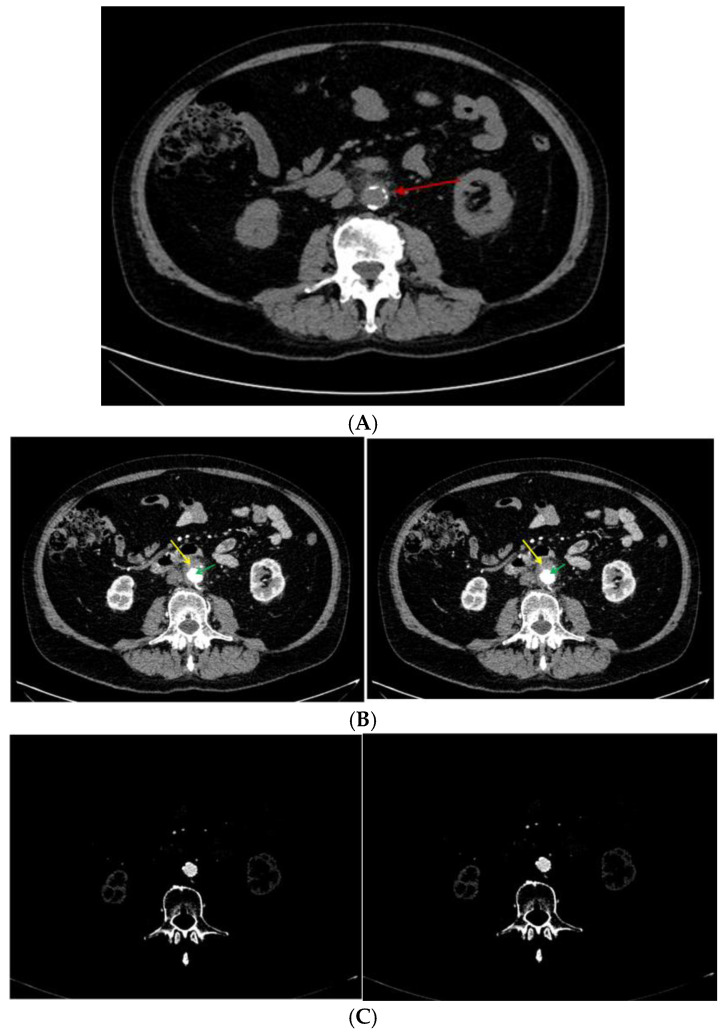
(**A**) Native abdominal CT scan revealing abdominal aorta calcifications and crescent-shaped thickening of the aortic wall (red arrow). (**B**) Abdominal angiography CT (green arrow = aorta; yellow arrow = parietal fibrosis) at the same level as seen in (**A**); parietal fibrosis (2/3 of the aortic circumference) was of 13 mm (a length of 27 mm, without hematic extravasation); captured at different levels, these parietal changes might explain the parietal resistance until the moment of emergency. Parietal fibrosis was considered amid the context of chronic inflammation underlying rheumatologic and cardiovascular ailments. (**C**) Angiography-CT (sections at the same levels).

**Figure 8 diagnostics-15-00144-f008:**
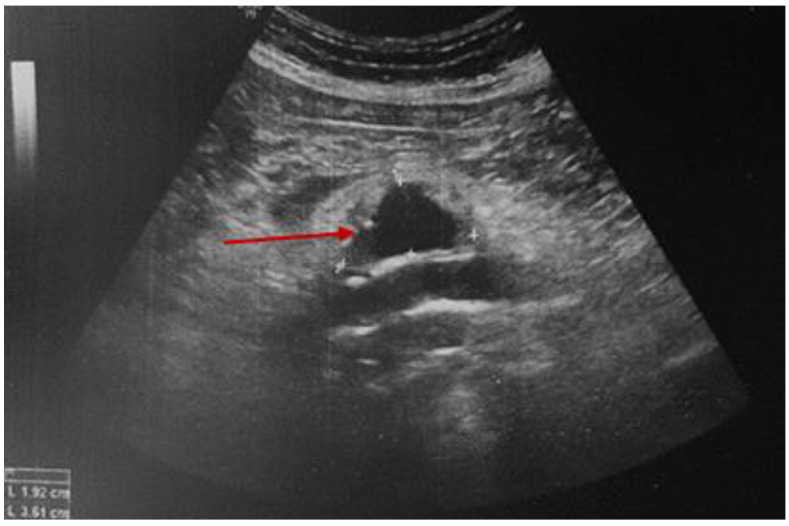
Abdominal ultrasound revealing a hypoechoic para-aortic area of 3.6 by 1.9 cm, embedded by the hyperechoic inflammatory tissue.

**Figure 9 diagnostics-15-00144-f009:**
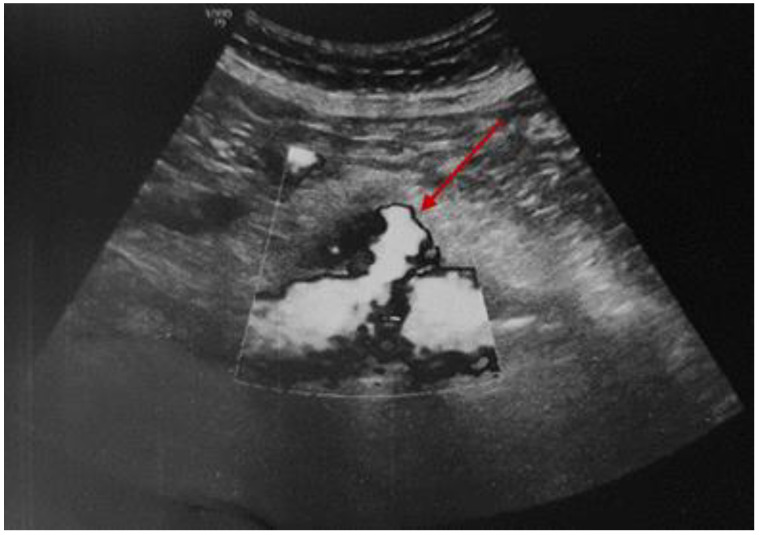
Abdominal ultrasound with color Doppler analysis showing the area of the blood extravasation (red arrow).

**Figure 10 diagnostics-15-00144-f010:**
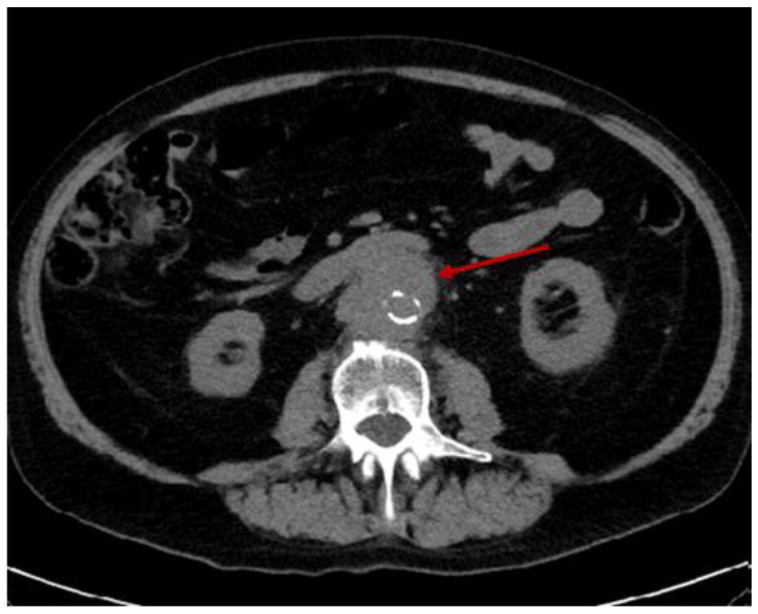
Native abdominal CT showing the enlargement of the peri-aortic area consistent with the diagnosis of a hematoma. This second CT scan was performed without a contrast medium, which was due to the patient’s chronic renal failure.

**Figure 11 diagnostics-15-00144-f011:**
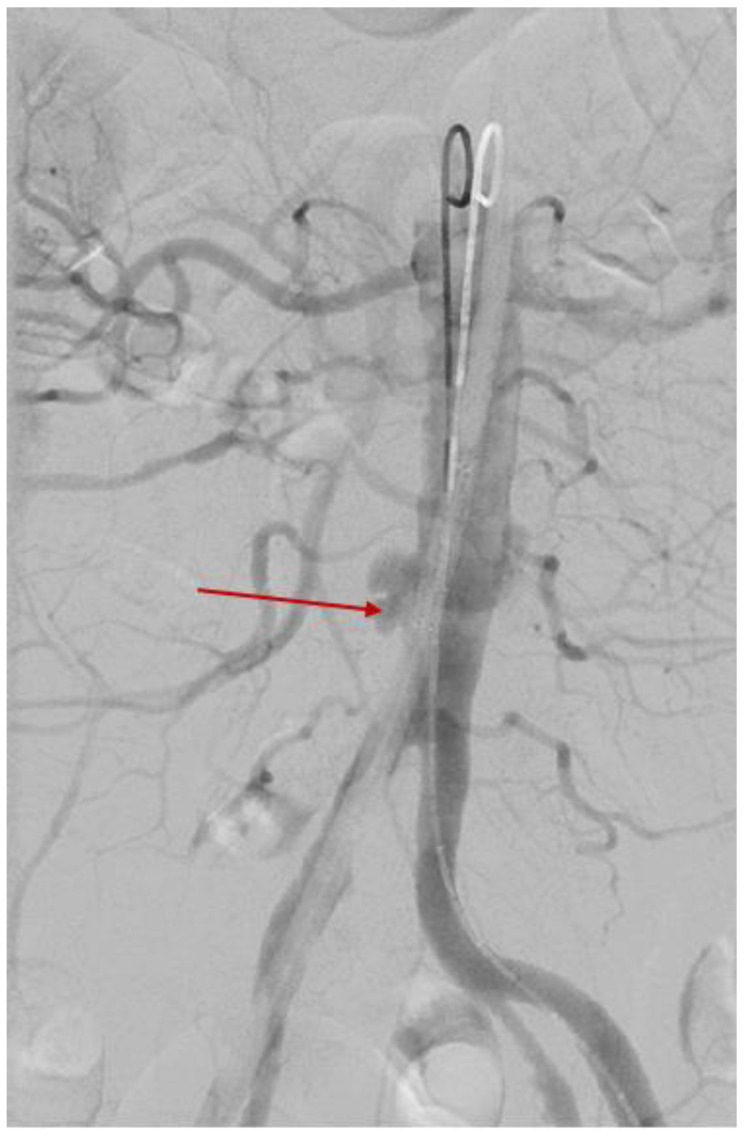
Pre-interventional arteriography showing the aortic lesion in the distal part of the abdominal aorta that formed an irregularly shaped pseudo-aneurysm (red arrow); the aorta was catheterized by placing a superstiff guidewire at this level. The procedure was followed by the implantation of 2 aortic stents of 2.4 × 4 cm.

**Figure 12 diagnostics-15-00144-f012:**
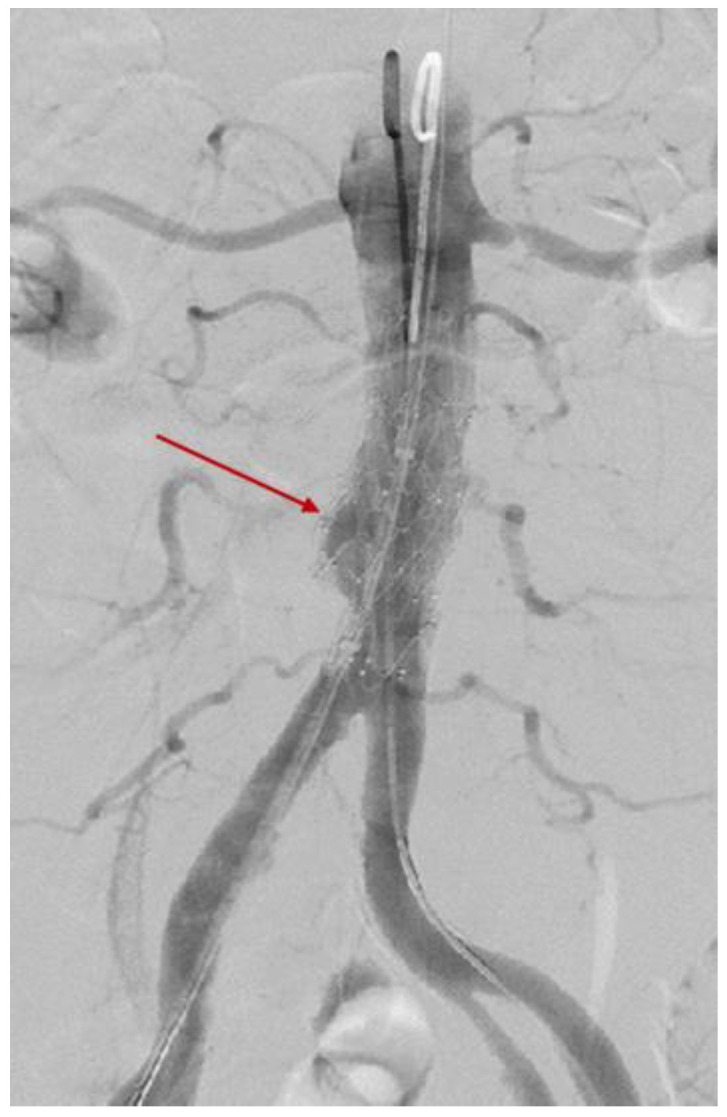
Stent dilatation with a 40 × 40 mm AndraTec balloon; the final result was optimal, without the extravasation of the contrast substance and restoration of the vascular axis. The two endovascular stents and the control injection showed a complete occlusion of the aortic rupture and the absence of the pseudo-aneurysm loading.

**Figure 13 diagnostics-15-00144-f013:**
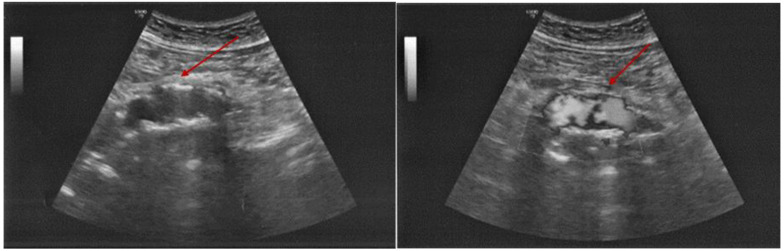
Successful vascular axis reconstruction (**left capture**) with complete permeability and normal anterograde blood-flow on Doppler examination (**right capture**).

**Table 1 diagnostics-15-00144-t001:** GPA and aortic involvement (the display starts with the most recent publication date) [46,47,48,49,50,51,52,53,54,55,56,57,58,59,60,61,62,63] (Abbreviations: AZA = azathioprine; cANCA = cytoplasmic-staining anti-neutrophil cytoplasmic antibodies; CYC = cyclophosphamide; ENT = ear, nose, and throat; GC = glucocorticoids therapy; MTX = methotrexate; NA = not available; PR3-ANCA = anti-proteinase-3-anti-neutrophil cytoplasmic antibody; PT = pharmacological treatment; RTX = rituximab; UFH = unfractionated heparin; W = warfarin).

First Author Year of Publication Reference Number	Patient’ Age (Years) Sex	Features of the Aortic Involvement Associated Symptoms/Signs	Granulomatosis with Polyangiitis			Outcome of the Aortic Lesion
			Disease-Related Involvement	ANCA Profile	Treatment	
Tzanninis 2022 [46]	58 M	Ascending aorta: aortitis and thrombus Chest pain, vomiting, and hemoptysis	Migratory arthritis Episcleritis	PR3-ANCA (74 UI/mL)	PT: UFH, W, GC, and CYC	Complete resolution of ascending aorta thrombus lesion
Rodriguez- Padilla 2021 [47]	74 M	Thoracic aorta: irregular peri-aortic mass surrounding the aortic graft Fever, epistaxis, cough, hemoptysis, and weight loss	Aorto-aortic bypass	cANCA (>1/20) PR3-ANCA (33 UI/mL)	PT: GC and MTX	Dimensional reduction of peri-aortic mass
Hesford 2021 [48]	61 F	Aortic arch: infiltrative wall process Thoracic back pain, progressive dyspnea, dry cough, and night sweats	Pulmonary infiltrates Chronic nasal congestion Hilar lymphadenopathy	PR3-ANCA (60–90 UI/mL)	PT: GC, CYC, and RTX	Clinical, serological, and radiological remission
Bernal 2019 [49]	34 M	Ascending aorta: circumferential wall thickening Facial swelling, nasal congestion, epistaxis, and progressive vision and weight loss	ENT Necrotizing scleritis Pan-uveitis	cANCA (>1/4) PR3-ANCA (>8 UI/mL)	PT: GC, RTX, and MTX	Significant decrease in the wall thickening around the ascending aorta + improvement in vision from the left eye, but no change from the right
Pan 2019 [50]	28 M	Thoracic aorta: dissection and aortic hematoma Aortitis Chest pain	Scleritis Conjunctivitis Arthritis	PR3-ANCA (180 UI/mL) MPO-ANCA (10 UI/mL)	PT: GC and CYC	Stable condition upon surgery: ascending aorta and arch replacement
Parperis 2019 [51]	71 F	Ascending aorta + aortic arch: thickening of the aortic wall Headache	Left eye blindness	pANCA (159 UI/mL)	PT: GC and MTX	Clinical and serological regression
Kim 2018 [52]	58 M	Ascending aorta: eccentric thickening of the aortic wall Mid-sternal pain, fever, and cough	Pulmonary infiltrates Hilar lymphadenopathy Scleral keratitis Pituitary adenoma resection	cANCA positive	PT: GC	Dimensional reduction in peri-aortic mass and pulmonary nodule
Revilla 2016 [53]	74 M	Abdominal aorta: soft tissue mass around the infrarenal aorta Back pain	Pulmonary infiltrates Pleural effusion Aorto-bifemoral bypass	cANCA (>1/20) PR3-ANCA (38 UI/mL)	PT: GC and MTX	Dimensional reduction in peri-aortic mass and pulmonary infiltrates
Ozaki 2015 [54]	60 F	Aortic arch + abdominal aorta: wall thickening Fever, epistaxis, and nasal obstruction	ENT Pulmonary infiltrates with cavities Skin ulcers	PR3-ANCA (153 UI/mL)	PT: GC, CYC, RTX, and AZA	Clinical, serological, and radiological remission
Ohta 2013 [55]	38 M	Thoracic aorta: dissection and rupture of aortic aneurysm Chest and back pain and loss of consciousness	ENT Glomerulonephritis	cANCA (x128)	PT: GC	Clinical and radiological remission upon surgery (J-graft insertion)
Amos 2012 [56]	64 M	Aortic arch + abdominal aorta: circumferential wall thickening Fever, malaise, dysuria, hematuria, and intermittent chest pain	Glomerulonephritis Diffuse alveolar hemorrhage	PR3-ANCA (55 UI/mL)	PT: GC, CYC, and MTX Plasma exchanges Hemodialysis	Slow regression of clinical manifestations
Shmagel 2011 [57]	68 F	Abdominal aorta: aneurysm and soft tissue mass around the infrarenal aorta Low abdominal pain	ENT Respiratory failure	cANCA (>1/20) PR3-ANCA (>100 UI/mL)	PT: GC, CYC, and MTX	Dimensional reduction of peri-aortic mass
Unlü 2011 [58]	43 M	Abdominal aorta: aneurysm and soft tissue mass around the infrarenal aorta Abdominal pain and generalized malaise	ENT nasopharyngeal ulceration Glomerulonephritis	NA	PT: GC	Clinical and radiological remission (Surgery: aorto-bi-iliac Dacron graft)
Minnee 2009 [59]	51 M	Abdominal aorta: aneurysm of the distal part of the aorta Low back pain and weakness of the upper and lower extremities with sensory loss	Testis The peripheral nerve system Skin	PR3-ANCA (>530 kU/L)	PT: GC, CYC, and Iloprost	Clinical remission
Carels 2004 [60]	63 M	Abdominal aorta: aneurysm of the distal part of the aorta Low back pain and paresthesia in the lower limbs	Lungs Bowel Polyneuropathy in the lower limbs	pANCA—positive	PT: GC	Clinical and serological remission (Surgery: aorto-bi-iliac Dacron graft)
Schildhaus 2002 [61]	63 M	Thoracic aorta: inflammatory lesions Weight loss, dyspnea, peripheral edema, and arthralgias	Skin	cANCA (1:320) PR3-ANCA (>100 U/mL)	Conservative treatment	Death: circulatory collapse
Blockmans 2000 [62]	42 M	Abdominal aorta: peri-aortitis and intramural dissection Abdominal pain	ENT Lungs Arthralgia Muscle weakness Paranesthesia	cANCA (1/1280) PR3-ANCA (55 UI/mL)	PT: GC and CYC	Slow recovery upon surgery: aorto-iliac graft + re-implantation of inferior mesenteric artery
Fink 1994 [63]	45 M	Abdominal aorta: wall thickening around the aorta, extending to the right iliac artery Intermittent right abdominal pain, malaise	ENT Lungs	cANCA positive	PT: GC, CYC	Good

## Data Availability

The research data that support the findings of this case are not publicly available. Other medical records are available upon request in accordance with the hospital rules, patient consent, and the local Ethics Committee.

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
