# Peer review of "Minimally Invasive Surgical Approach in Granulomatosis with Polyangiitis Complicated with Intramural Descending Aorta Hematoma Followed by Aortic Wall Rupture"

_diagnostics, 2025, doi:10.3390/diagnostics15020144_

Round 1

Reviewer 1 Report

Comments and Suggestions for Authors

Dear Authors,

thank you for this nice reports

Please find my review comments: 

Introduction is too long. I suggest to delete from line 93 to 114 (this part is not interesting make the paper difficult to read)

Methods: 

for mesh terms: did you check also ANCA vasculitis and Aorta? 

Medical history of the patients: 

Please give more details what was the titer of anca (page 4 of 21)

What was the kidney parameters (creatinine, proteinuria, hematuria....) (page 5 of 21)

Page 6 of 21 (line 195): the relapse occur after treatment withdrawal? 

Current hospitalisation:

Page 6 of 21 (line 218): 125 mg/dl or 125 mg/L??

Page 7 of 21: i think a figure showing evolution of ANCA titers is needed (from first diagnosis, first relapse to the current hospitalisation). Since yu do not have any pathological exam of the aorta, i think it is necessary to show the evolution of ANCA titers

Author Response

Response to Review 1 Comments

Dear Reviewer,

Thank you very much for your time and your effort to review our manuscript.

We are very grateful for providing your valuable feedback on the article.

Here is our response and related amendment that has been made in the manuscript according to your review (marked in yellow color).

Dear Authors,

Thank you for this nice report.

Please find my review comments: 

  1. Introductionis too long. I suggest to delete from line 93 to 114 (this part is not interesting make the paper difficult to read)

Thank you very much. According to your recommendation, we removed those lines. Thank you.

  1. Methods:  for mesh terms: did you check also ANCA vasculitis and Aorta? 

Thank you very much. We limited the search to “granulomatosis with polyangiitis (alternatively, Wegener’s granulomatosis’)” and “aorta” in order to reflect the data of the present case. Thank you

  1. Medical history of the patients:  Please give more details what was the titer of anca (page 4 of 21)

Thank you very much. According to your recommendation, we provided ANCA titers during follow-up (Adnexa 1).

  1. What was the kidney parameters (creatinine, proteinuria, hematuria....) (page 5 of 21)

Thank you very much. According to your recommendation, we provided the lab assessments on current admission (Adnexa 1)

  1. Page 6 of 21 (line 195): the relapse occured after treatment withdrawal? 

Thank you very much. We confirmed and added it. “The recurrence with pulmonary manifestations occurred approximately 5 years after the discontinuation of oral cyclophosphamide therapy (which had been maintained for about 2 years). At that time, the patient was on low-dose maintenance corticosteroid therapy (7.5–10 mg/day). Subsequently, azathioprine was introduced as maintenance remission therapy.” Thank you

  1. Current hospitalisation: Page 6 of 21 (line 218): 125 mg/dl or 125 mg/L??

Thank you very much. According to your recommendation, we corrected it (“mg/L”). Thank you

  1. Page 7 of 21:I think a figure showing evolution of ANCA titers is needed (from first diagnosis, first relapse to the current hospitalisation). Since you do not have any pathological exam of the aorta, I think it is necessary to show the evolution of ANCA titers.

Thank you very much. We provided a detailed table with respect to ANCA titers and their evolution that pinpoints the disease-related status. Thank you

Thank you very much.

Reviewer 2 Report

Comments and Suggestions for Authors

Thank you for the opportunity to review this case report that presents an interesting case. However, many important details are missing from the text. Therefore, the authors should include the sex of the patient and laboratory values in the table. They should not include photos but original images in high-resolution quality (150-300 DPI). The acronym CT for computed tomography should be used, and CT-angiography images should be added, not just the basal abdominal CT. The authors should specify the type of aortic dissection (abdominal? Type B associated with GPA?) and include CT  angiography images of the dissection. They should also report the symptomatic and conservative treatment of the dissection. Why was an Aortic endoprosthesis not performed before the aortic rupture? Additionally, the authors should include CT angiography images in Fig 10.

Author Response

Response to Review 2 Comments

Dear Reviewer,

Thank you very much for your time and your effort to review our manuscript.

We are very grateful for your insightful comments and observations, also, for providing your valuable feedback on the article.

Here is a point-by-point response and related amendments that have been made in the manuscript according to your review (marked in yellow color).

  1. Thank you for the opportunity to review this case report that presents an interesting case. However, many important details are missing from the text.

Thank you very much. We addressed them point by point.

  1. Therefore, the authors should include the sex of the patient and laboratory values in the table.

Thank you very much. According to your recommendation, we specified these data. “This was a 64-year-old male patient with an 18-year history of GPA who presented progressively increased lumbar pain over the last month.” In Adnexa 1, we provided the lab assessments. Thank you

  1. They should not include photos but original images in high-resolution quality (150-300 DPI). The acronym CT for computed tomography should be used, and CT-angiography images should be added, not just the basal abdominal CT.

Thank you very much. According to your recommendation, we introduced the abbreviation “CT”.  We introduced all available captures to the best we could and to the best that we were allowed in order to be reproduced them within an article with respect to the imaging evaluation. Of note, at the moment (during follow-up) of aortic rupture, a native CT scan was done as an emergency and then the patient was immediately referred to the surgery room/department. Thank you

  1. The authors should specify the type of aortic dissection (abdominal? Type B associated with GPA?) and include CT  angiography images of the dissection.

Thank you very much. There was no typical aortic dissection. The interventional radiology specialist noted at that point in his report a rupture/transection of the aortic wall, an injury that was initially stabilized by the fibrous sheath (most likely due to pre-existing inflammation) around the aorta. We already explained the limits of our imaging captures. Thank you

  1. They should also report the symptomatic and conservative treatment of the dissection.

Thank you very much. According to your recommendation, we added it: “The patient initially received analgesic treatment with acetaminophen and metamizole sodium, but with partial relief of symptoms that required the subsequent escalation of therapy to nefopam hydrochloride and tramadol. Also, special attention was directed to achieve strict blood pressure control.” Thank you

  1. Why was an Aortic endoprosthesis not performed before the aortic rupture?

Thank you very much. Aortic endoprosthesis was not performed because there was no imaging suspicion of dissection at that point. Thank you

  1. Additionally, the authors should include CT angiography images in Fig 10.

Thank you very much. We already explained the limits with respect to the availability of such captures. Thank you

Thank you very much

Round 2

Reviewer 2 Report

Comments and Suggestions for Authors

Thank you for the revised version. However, the angiography CT images are necessary. On page 222, line 223, it is reported that a CT with contrast administration was performed, therefore enhanced CT images should be included. In order to demonstrate the parietal fibrosis, authors should include the enhanced CT images, and to show the intramural aortic hematoma, authors should also add the angiography CT images. It is impossible to diagnose aortic emergencies without angiographic CT acquisition. Please include them, as the acquisition of only basal CT would be a methodology error. I suggest adding radiologists in your work to solve the problem of images but they are important to understand the clinical case and for the scientific validity. Thank you for the other corrections.

Author Response

Response to Review 2 Comments – second round

Dear Reviewer,

Thank you very much for your time and your effort to review our manuscript.

We are very grateful for providing your valuable feedback on the article.

Here is our response and related amendment that has been made in the manuscript according to your review (green font).

Thank you for the revised version. However, the angiography CT images are necessary. On page 222, line 223, it is reported that a CT with contrast administration was performed, therefore enhanced CT images should be included.  In order to demonstrate the parietal fibrosis, authors should include the enhanced CT images, and to show the intramural aortic hematoma, authors should also add the angiography CT images. It is impossible to diagnose aortic emergencies without angiographic CT acquisition. Please include them, as the acquisition of only basal CT would be a methodology error. I suggest adding radiologists in your work to solve the problem of images but they are important to understand the clinical case and for the scientific validity. Thank you for the other corrections.

Thank you very much. We included all available captures at Figure 7 that involve the contrast abdominal CT scan, including angiography CT images. This is at the starting point of the 12-day surveillance period when the patient was not considered an emergency, but remained under close surveillance and symptomatic medical therapy amid hospitalization. The parietal fibrosis was considered the clue of resistance until the endovascular intervention.

We provided as Adnexa 2 the timeline of imaging assessment according to the case on point.

No additional contrast was added to the emergency (native) CT scan (Figure 10) at that time since the patient was referred as emergency to the interventional arteriography (with pre-interventional captures being already provided) and an overexposure to contrast was considered unnecessary due to his chronic renal disease and renal insufficiency under these circumstances of imminent shock/collapse. Pre-interventional arteriography highlighted the abdominal aorta rupture and the formation of a pseudoaneurysm as shown in Figure 11. Prior to this emergency point, the additional description was already provided in the captures from Figures number 8-9, meaning ultrasound follow-up until the moment of emergency,

Notably, this is real-life medicine and we stand by all the presented data and associated captures. Exceptional aspects are part of this case report amid a multidisciplinary, emergency, military hospital, including the successful minimally invasive intervention.

All the contributors who are qualified as authors are already displayed.

Thank you very much.

Round 3

Reviewer 2 Report

Comments and Suggestions for Authors

Thank you for adding the pictures of the angiography CT. However, there are now too many and they are unnecessary. The authors should include a single angiography CT image and place it close to the baseline, creating figures a (basal) and b (angiography CT) at the same level. Additionally, the authors should explain in the text why the second CT scan was performed without contrast medium, which was due to the patient's chronic renal failure.

Author Response

Response to Review 2 Comments – third round

Dear Reviewer,

Thank you very much for your time and your effort to review our manuscript.

We are very grateful for providing your valuable feedback on the article.

Here is our response and related amendment that has been made in the manuscript according to your review.

Thank you for adding the pictures of the angiography CT. However, there are now too many and they are unnecessary. The authors should include a single angiography CT image and place it close to the baseline, creating figures a (basal) and b (angiography CT) at the same level.

Thank you very much. According to your recommendation, we selected and removed the additional images. Thank you

Additionally, the authors should explain in the text why the second CT scan was performed without contrast medium, which was due to the patient's chronic renal failure.

Thank you very much. According to your recommendation, we added this information. Thank you

Thank you very much.